# Prognostic Factors and Long-Term Survival in Locally Advanced NSCLC with Pathological Complete Response after Surgical Resection Following Neoadjuvant Therapy

**DOI:** 10.3390/cancers12123572

**Published:** 2020-11-30

**Authors:** Filippo Lococo, Carolina Sassorossi, Dania Nachira, Marco Chiappetta, Leonardo Petracca Ciavarella, Emanuele Vita, Luca Boldrini, Jessica Evangelista, Alfredo Cesario, Emilio Bria, Stefano Margaritora

**Affiliations:** 1Università Cattolica del Sacro Cuore, 00168 Rome, Italy; dania.nachira@policlinicogemelli.it (D.N.); stefano.margaritora@policlinicogemelli.it (S.M.); 2Thoracic Surgery, Fondazione Policlinico Universitario A. Gemelli IRCCS, 00168 Rome, Italy; sassorossi.caro@gmail.com (C.S.); marco.chiappetta@policlinicogemelli.it (M.C.); leonardo.petracca@gmail.com (L.P.C.); evangelistajessica664@gmail.com (J.E.); 3Comprehensive Cancer Center, Fondazione Policlinico Universitario Agostino Gemelli IRCCS, 00168 Roma, Italy; dr.emanuele.vita@gmail.com (E.V.); emilio.bria@policlinicogemelli.it (E.B.); 4Medical Oncology, Dipartimento di Medicina e Chirurgia Traslazionale, Università Cattolica del Sacro Cuore, 00168 Roma, Italy; 5Radiotherapy Unit, Fondazione Policlinico Universitario A. Gemelli IRCCS, 00168 Rome, Italy; luca.boldrini@policlinicogemelli.it; 6Open Innovation Manager, Direzione Scientifica, Fondazione Policlinico Universitario A. Gemelli IRCCS, 00168 Rome, Italy; alfredo.cesario@policlinicogemelli.it

**Keywords:** NSCLC, induction therapy, pathological complete response

## Abstract

**Simple Summary:**

Higher response may be achieved with induction therapy (IT) and better survival results could be expected after complete surgical resection for non-small-cell lung cancer (NSCLC) patients. Thus, locally advanced (LA)-NSCLC patients with pathological complete response (pCR) are optimal candidates to undergo surgery after IT, achieving good to very good long-term survival. Herein, we performed a retrospective analysis on a large cohort of locally advanced NSCLC patients who achieved pCR after IT and surgery, exploring long-term survival and factors affecting prognosis. We observed a rewarding 5-year overall survival (56%) with baseline N2 single-station disease and adjuvant therapy after surgery associated with better prognosis. These findings may be useful to better define the strategy of care in this highly selected subset of NSCLC patients.

**Abstract:**

*Background:* Outcomes for locally advanced NSCLC with pathological complete response (pCR), i.e., pT0N0 after induction chemoradiotherapy (IT), have been seldom investigated. Herein, long-term results, in this highly selected group of patients, have been evaluated with the aim to identify prognostic predictive factors. *Methods:* Patients affected by locally advanced NSCLC (cT1-T4/N0-2/M0) who underwent IT, possibly following surgery, from January 1992 to December 2019, were considered for this retrospective analysis. Survival rates and prognostic factors have been studied with Kaplan-Meier analysis, log-rank and Cox regression analysis. *Results:* Three-hundred and forty-three consecutive patients underwent IT in the considered period. Out of them, 279 were addressed to surgery; among them, pCR has been observed in 62 patients (18% of the total and 22% of the operated patients). In the pCR-group, clinical staging was IIb in 3 (5%) patients, IIIa in 28 (45%) patients and IIIb in 31 (50%). Surgery consisted of (bi)lobectomy in the majority of cases (80.7%), followed by pneumonectomy (19.3%). Adjuvant therapy was administered in 33 (53.2%) patients. Five-year overall survival and disease-free survival have been respectively 56.18% and 48.84%. The relative risk of death, observed with the Cox regression analysis, was 4.4 times higher (95% confidence interval (CI): 1.632–11.695, *p* = 0.03) for patients with N2 multi-station disease, 2.6 times higher (95% CI: 1.066–6.407, *p* = 0.036) for patients treated with pneumonectomy and 3 times higher (95% CI: 1.302–6.809, *p* = 0.01) for patients who did not receive adjuvant therapy. *Conclusions:* Rewarding long-term results could be expected in locally advanced NSCLC patients with pCR after IT followed by surgery. Baseline N2 single-station disease and adjuvant therapy after surgery seem to be associated with better prognosis, while pneumonectomy is associated with poorer outcomes.

## 1. Introduction

Lung cancer remains the leading cause of cancer mortality worldwide, accounting for 1.8 million new diagnoses per year (13% of all diagnosed cancers) [1]. Non-small-cell lung cancer (NSCLC) represents more than 80% of total lung cancer cases, of which 19% are early-stage, 25% are locally advanced and 56% present with distant metastases ab initio [2].

While survival has increased in the last decades and guidelines substantially agree on the therapeutic pathway for early-stage lung cancer, the strategy of care for locally advanced (LA) NSCLC (mostly N2 disease) has not been adequately standardized and survival outcomes are so far almost disappointing, with no remarkable improvements in the last 30 years.

Indeed, the choice of local treatment modalities can vary across countries and centers [3,4].

In the current National Comprehensive Cancer Network (NCCN) evidence-based clinical practice guidelines (version 3 February 2020), the expert panel state that “surgery may be appropriate for selected N2 disease, especially those whose disease responds to induction chemotherapy” [2].

Despite that the real benefit of induction therapy (IT) (chemotherapy or radio-chemotherapy) in LA-NSCLC is far from being defined [5], its application in daily clinical practice is widely adopted both in and outside Europe, with some centers (as ours) having long-term experience with this approach. Indeed, from a theoretical point of view, induction treatment in LA-NSCLC should increase resectability by downstaging lung cancers, reduce the resection extent and both local and distant recurrence rate controlling microscopic distant metastatic spread, and finally increase overall survival rates.

Surgery is generally the most accepted treatment after IT in patients with complete or partial response to IT itself or those with stable disease and evidence of technical feasibility of resection with radical intent [5,6,7].

Indeed, consistent evidence suggests that downstaging after IT is directly associated with an improvement of disease-free survival (DFS) and lower distant recurrence rates [8].

Fundamentally, the higher the IT response, the better survival results are expected after complete surgical resection. Thus, LA-NSCLC patients with pathological complete response (pCR) are therefore optimal candidates to undergo surgery after IT, achieving good to very good long-term survival [9,10,11].

However, this specific subset of patients is highly selective and only 10–30% of LA-NSCLC patients present with pCR after IT according to literature data [9], with no prognostic factors validated in this sub-population. Moreover, the administration of adjuvant therapy after surgery in pCR patients is still the object of debate as no focused analyses on this topic are available. 

Furthermore, in the last update of the NCCN guidelines, the expert panel deleted the recommendation for post-operative chemotherapy in patients with N2 disease receiving IT, because no evidence was available on this specific issue.

The aim of this retrospective study is to perform a comprehensive analysis of the clinical and pathological characteristics of a large patient cohort (20-year experience) undergoing pCR after surgical resection following IT.

Herein, data on long-term results in terms of 5-year survival (LTS) and information on prognostic factors are described in detail and discussed.

### 2. Materials and Methods

A retrospective review of a prospective institutional lung cancer database was conducted to select the clinical records of LA-NSCLC (cT1-T4/N0-2/M0) who achieved pCR at surgery after IT.

From January 1992 to December 2019, 343 consecutive LA-NSCLC cases underwent IT and out of them, 279 were submitted to surgical resection after favorable clinical restaging. Of these, 62 patients (18% of the total sample and 22% of the operated patients) showed pCR after surgery and were therefore retrospectively selected for this analysis.

Institutional review board approval had been preliminarily obtained for the use of data stemming from standard clinical practice for research purposes, as no additional interventions were planned (retrospective observational study).

Based on the information available from clinical records, demographic and clinical features were collected and taken into consideration for statistical analysis.

Follow-up data were retrieved from our database or, when absent or corrupted, by direct phone interview (with privacy-related issues being covered by the original comprehensive informed consent) with patients or, in the case of death, with legal representatives from the family (next of kin). All patients had at least a 6-month follow-up at the time of the analysis.

The 8th edition of the American Joint Committee on Cancer tumor (T), node (N) and metastasis (M) staging was adopted [12]. Inclusion criteria to the multimodal treatment were previously reported [9,13].

Pre-treatment evaluation included patient history, physical examination, lung function tests, complete blood chemistry, computed tomography (CT) scan of the chest, brain and upper abdomen, brain magnetic resonance imaging (MRI), 18Fluoro-D-Glucose Positron Emission Tomography (18F-FDG PET), bone radionuclide scan and fiberotic bronchoscopy. NSCLC was diagnosed based on pathological examination of the material obtained via endo-bronchial biopsy or CT-guided fine-needle aspiration.

In detail, clinical staging was based on radiological (CT-scan and MRI) results and radiometabolic findings (18F-FDG PET/CT, in NSCLC cases observed after 2008) but the mediastinal involvement was always pathologically proven by endoscopy (trans-bronchial/trans-esophageal) or via collar mediastinoscopy or surgical procedures (Chamberlain’s mediastinotomy or diagnostic thoracoscopy).

Pathological complete response was defined as the absence of tumor cells in all surgical specimens (ypT0N0).

### 2.1. Induction Therapy

Procedures and schedules regarding the IT protocols are extensively described in Trodella et al. [7]. In addition to minor changes in drug selection, uniformly platinum-based compounds (CBDCA, CDDP) plus 5 fluorouracil (5FU) until Gemcitabine (GEM) in substitution of 5 FU was introduced over the years, and there has been an evolution of the used irradiation technique from 2- to 3-dimensional conformal radiotherapy (CRT) to intensity modulated radiation therapy (IMRT) in 2002 and volumetric modulated arc therapy (VMAT) more recently. 

The total administered rditrapy (RT) dose was 50.4 Gy in all cases, with classical (1.8 Gy per fraction) or a hyper-fractionated schedule (640 cGy at 40 cGy fractions bis in die 1–8 q14 followed by 50.4 Gy at 1.8 Gy per fraction). 

In a minor proportion of cases (see Table 1), an IT protocol consisted of chemotherapy only. In detail, three cycles of preoperative chemotherapy (cisplatin plus gemcitabine) were administered (in line with Scagliotti and co-workers [14]) followed by surgery. A complete clinical and radiological restaging was performed 4–6 weeks after the end of IT. 

### 2.2. Restaging

A complete clinical and radiological re-evaluation was performed 4 weeks after the treatment end. Clinical restaging was realized with CT scan, and with 18F-FDG PET/CT (cases observed after 2008). All NSCLC patients with no progression at restaging and with a resectable disease were addressed to surgery after careful cardio-pulmonary evaluation.

We did not perform re-mediastinoscopy. The multidisciplinary thoracic oncology team reevaluated response to induction treatment.

Among pCR patients who underwent PET/CT scan (#28 cases), persistence of a significant uptake at the level of lung lesion (#19 cases) or mediastinal lymph nodes (#4 cases) was detected.

### 2.3. Surgery

When indicated, surgery was performed on average 16 (range 10–21) days after restaging. Considering the initial locally advanced disease stage, a parenchymal resection to an extent less than a lobectomy was considered as oncological-inappropriate and therefore never performed. Systematic mediastinal lymph node dissection was performed following the principles reported in the European Society of Thoracic Surgery (ESTS) guidelines [15]. At surgery, we removed, at least, three mediastinal nodes from three stations in which the subcarinal is always included. Pathologic evaluation includes all lymph nodes resected separately and those remaining in the lung specimen.

Pathological complete response (pCR) was defined as absence of viable tumor cells in the surgical specimen: a final pathological stage 0 (pT0 N0 M0) was then assigned. Patients with only residual microscopic foci of disease were considered to have a pT1 disease and were therefore excluded from the study.

### 2.4. Adjuvant Therapy and Surveillance Protocol

After surgery, consolidation chemotherapy was performed in about 30% of cases while radiation therapy was performed in 11 patients where no RT was administered in the neoadjuvant setting.

Surveillance after surgery was scheduled every 3–6 months for the first 3 years, then every 6 months for another 2 years and finally every year thereafter. The surveillance protocol consisted of physical examination, chest CT-scan (along with brain CT-scan and abdominal ultrasound in the first 3 years). 18F-FDG PET/CT was not routinely performed in this setting.

### 2.5. Statistical Analysis

Patients’ characteristics were described by mean, standard deviation (SD), median, minimum and maximum for continuous variables, and frequencies and percentages for categorical ones.

Time-to-event analysis, at 3 and 5 years from surgery, was performed on these outcomes by means of Kaplan-Meier survival curves and Log-Rank tests. When the upper bound of the 95% confidence interval (CI) lay outside the observed timeframe, the last time of follow-up was reported. Multivariate analysis was performed with the Cox proportional hazards regression model to assess the prognostic role of several variables on patients’ survival and recurrence rate. For all tests, *p*-value < 0.05 was considered as statistically significant. Statistical analyses were performed using IBM SPSS Statistics for Windows (version 25.013, IBM Corp, Armonk, NY, USA) and STATA software package (STATA/SE version 10.0, Stata Corp., College Station, TX, USA).

## 3. Results

A consort-type diagram (Figure 1) reported the selection process of the study group (62 pCR-NSCLC patients corresponding to 18% of all considered patients and 22% of those who underwent surgery).

The main clinical and pathological results, surgical notes and follow-up information are summarized in Table 1.

Mean age and male/female ratio were 62.44 ± 8.76 years and 57/5, respectively. As reported in Table 1, the baseline clinical stage was IIb in 3 (5%) patients, IIIa in 28 (45%) patients and IIIb in 31 (50%). Single-station involvement was detected in most of the baseline N2 patients (31/56 patients, 58.5%), while multi-station N2 involvement was observed in the remaining cases. Induction therapy consisted of combined radio-chemotherapy in the large majority of cases (82.3%). After surgery (see Table 1), adjuvant therapy was administered in 33 (53%) patients, according to the discretion of the treating medical oncologist. Adjuvant chemotherapy consisted of platinum-based protocols; in detail, 18 patients received docetaxel and cisplatin, 11 gemcitabine and cisplatin and 4 gemcitabine and carboplatin combinations.

### 3.1. Pattern of Recurrence after Surgery

Recurrence were observed in 32 patients during the follow-up period (overall recurrence rate (RR) 51.6%). The disease relapsed more often in a distant site (23 patients, distant recurrent rate (DRR) 37.1%) than locally (9 patients, local recurrent rate (LRR) 14.5%). The most common site of relapse was in the brain, followed by adrenal gland and liver (see Table 2). Interestingly, patients who underwent pneumonectomy presented a different pattern of recurrence after surgery compared with patients who underwent lobectomy or bilobectomy (see Table 2 for details).

Indeed, while the overall RR was similar (50% after pneumonectomy vs. 52% after (bi)lobectomy), the LRR was substantially different (LRR: 0% vs. 18%). The pattern of recurrence was also influenced by the administration of adjuvant therapy. More precisely, in 8/33 (24.2%) patients who underwent adjuvant therapy, distant relapse occurred, while recurrences occurred in 15/29 (51.7%) patients who did not receive systemic treatment after surgery.

### 3.2. Long-Term Outcomes and Prognostic Factors

The overall mean (±standard deviation) follow-up duration was 56.2 ± 42.3 months. The estimated median survival time was 77.0 ± 14.8 months (95% CI: 48.0–106.0), while median time to recurrence was 60.0 ± 19.6 months (95% CI: 21.6–98.4). Three- and five-year LTS rates for the entire cohort of patients were 60% (95% CI: 49–80%) and 56% (95% CI: 46–77%) respectively, as reported in Figure 2A. Similarly, 3- and 5-year DFS rates were 54% (95% CI: 50–81%) and 45% (95% CI: 43–76%) (Figure 2B).

The observed 3- and 5-year DFS rates were 57% (95% CI: 50–81%) and 46% (95% CI: 43–76%) respectively (Figure 2C), while the 3- and 5-year distant DFS rates were 57% (95% CI: 50–81%) and 53% (95% CI: 43–76%), respectively (Figure 2D). Survival rates were calculated according to clinical, surgical, pathological and post-operative variables (results are summarized in Table 3).

Log-rank analysis showed that a better outcome could be expected in patients with initial N2 single-level involvement if compared with patients with initial N2 multiple-level involvement in terms of mean survival (160.2 ± 21.3 months vs. 46.2 ± 5.8 months), 5-years LTS (80% vs. 32%, *p* < 0.001) and DFS (69% vs. 19%, *p* < 0.001), as reported in Figure 3. A significant difference was found also concerning the type of resection. Long-term survival in patients who underwent pneumonectomy was significantly lower than that observed in the lobectomy group (median survival 37.0 ± 27.9 vs. 86.0 ± 15.6; 5-year LTS 33% vs. 61%, *p* = 0.024; DFS 29% vs. 48%, *p* = 0.220). 

Finally, the analysis revealed that the administration of adjuvant therapy positively influenced patients’ long-term prognosis (mean survival 115.6 ± 12.4 vs. 66.6 ± 15.7; LTS: 70% vs. 39%, *p* = 0.005; DFS: 57% vs. 29%, *p* = 0.013). No differences in long-term outcomes were found when comparing other selected factors (see Table 3). In detail, no significant differences in 5-year OS were found according to T-Stage, number of removed lymph nodes and clinical stage. As reported in Table 4, a 4.37 times higher relative risk of death (95% CI: 1.632–11.695, *p* = 0.003) has been estimated at the Cox regression analysis for patients with N2 multi-station disease. This risk was 2.61 times higher (95% CI: 1.066–6.407, *p* = 0.036) for patients treated with pneumonectomy and 2.98 times higher (95% CI: 1.302–6.809, *p* = 0.01) for the patients who did not receive any adjuvant therapy.

## 4. Discussion

Both surgery and other oncological treatments alone are not sufficient to control LA-NSCLC, which justifies preferring a combined approach with IT followed by surgery when clinically feasible, and especially when a mediastinal downstaging or a pCR has been achieved in the neoadjuvant setting [8]. From a theoretical point of view, the IT in LA-NSCLC should have the following purposes: (i) to increase resectability by downstaging lung cancer, (ii) to reduce the extent of resection, (iii) to reduce the local and distance recurrence rate, by controlling microscopic distant metastastic spread, and (iv) to increase the overall survival. Robust evidence suggests that better long-term outcomes could be expected after surgery in patients who presented with a downstaging after neoadjuvant therapy (i.e., mediastinal downstaging) [8,10,13]. Although it does not represent the primary goal of neoadjuvant therapy, one of its potential outcomes is the achievement of pathological complete response (defined as the absence of any residual neoplastic cell in the surgical specimen), which represents the best scenario with the highest survival probabilities. In particular, reviewing the largest clinical series reported in the literature, the rates of pCR for patients with LA-NSCLC treated with neoadjuvant therapy range between 8% and 45%, depending on the IT approach used [10,11,13,16,17,18]. Our data confirm from a long-term experience that pCR may be obtained in a significant proportion of patients treated with an IT protocol (22% of all patients who underwent surgery in our cohort) and this result may be achieved with acceptable treatment-related toxicity and surgical morbidity rates (see consort diagram in Figure 1). The high rate of pCR may be associated with the high proportion of patients who received radiation therapy (especially when high-dose treatment is administered) in IT protocols, probably due to a more effective loco-regional control, as also confirmed by the low LRR rates registered in our experience. The long-term survival observed in this subset of patients is rewarding (about 56% at 5 years), despite the fact that recurrences occurred in more than half of them. In particular, the pattern of failure revealed that the “Achille’s heel” of this multimodal approach relies in distant disease control, as clearly emerged observing the pattern of failure in Figure 2.

Based on our results, we aim to evaluate the efficacy of the recently introduced immunotherapy in a neoadjuvant setting, with the hope to improve the proportion of patients who achieved pCR and those with no distant recurrence after the multimodal approach. In this sense, a recent meta-analysis on the efficacy of immunotherapy in resectable NSCLCs reported encouraging data on 252 cases with pCR ranging from 4.9% to 27.3% [19]. Accordingly, it is reasonable to expect better results in terms of DFS and OS from future multimodal protocols, including immunotherapy. In this study, we have demonstrated several factors that appear to be associated with improved survival in patients affected by lung cancer achieving pCR. Firstly, we observed that initial single-level N2 disease is associated with improved OS. This factor was not previously explored in other series on pCR patients and seems to be a promising and easy-to-use prognostic factor in this subset of patients [10,16,17,18,20]. Other authors have reported better survival results in patients with persistent N2 disease after IT and followed by surgery, when a single N2 station is involved [21]. Similarly, we noted that among the N2-NSCLC patients achieving pCR, those who presented with baseline single N2 station may be considered the subgroup of patients with the better long-term OS. On the contrary, we found that pneumonectomy was also associated with decreased survival and this finding is in line with the experience of Martinez-Meehan et al. [18]. Such results deserve attention. 

Firstly, the complication rate after pneumonectomy following induction chemoradiotherapy is higher than lobectomy, but still acceptable. In detail, in a previous study performed by our team a few years ago [22], we reported a major complication rate of 14.3% in this subset of patients (in line with other similar series [23,24]). Therefore, the increased risk of death reported in the present study seems not associated to the consequence of the surgical procedure itself but is probably related to the baseline (more advanced) stage of these patients, who require a pneumonectomy for a complete resection. Moreover, when we look at the pattern of failure after pneumonectomy (Table 2), a better local control could be observed, while distant recurrences frequently occurred. In post-neoadjuvant pneumonectomy patients, the proportion of them undergoing adjuvant therapy is very low and this may explain: (1) the high rate of distant recurrences, and (2) the overall poor prognosis of these patients, confirming the protective role of adjuvant therapy also in pCR patients. Therefore, the detrimental effect of pneumonectomy could be related to the loss of the protective effect of adjuvant treatments in this subgroup. However, the opportunity to perform a pneumonectomy in these patients is almost debatable and not routinely recommended.

On a different aspect, an accurate analysis of the survival data matched with the pattern of failure after surgery may highlight some critical points in the strategy of care of the patients whose data have been analyzed. The overall local recurrence proportion is substantially acceptable (14.5%), while the distant control of the disease is quite poor (37%), with high rates of brain metastases (14 patients, 22.5%) observed in the overall cohort (see Table 2). It should be noted that none of these patients had been submitted to prophylactic cranial irradiation (PCI), even if based on the results of a large multicenter randomized trial, PCI has been suggested as a valuable tool to reduce the risk of brain metastases in LA-NSCLC after IT protocols [23]. Our data confirm the existence of a clinical rationale of this treatment even in pCR patients, but our results are limited and not robust enough to draw definitive conclusions on this issue. Concerning the improved survival in pCR patients treated with adjuvant therapy after surgery, the results reported in the literature are not univocal [10,16,17,18]. Indeed, while its role is substantially accepted in NSCLC patients with persistence of N2 disease after neoadjuvant therapy, the published experiences are contradictory in pCR patients [18]. Martinez-Meehan et al. recently reviewed the survival data of the National Cancer Database, focusing on 759 stage I-III NSCLC patients who achieved pCR after multimodal therapy, including surgery. Only 71 (9.3%) underwent adjuvant treatments and the authors did not observe a statistically significant difference in terms of long-term survival [18]. Similarly, Melek and colleagues reported a series of 72 pCR NSCLC patients where only 7 (nearly 10%) received adjuvant therapy, which did not influence the long-term outcome at univariate survival analysis [10]. On the contrary, Kawayake’s experience of 38 pCR patients, with 25 (65%) of them receiving adjuvant treatment, observed impressive (79.5%) 5-year OS rates in the entire cohort, suggesting a positive prognostic role of adjuvant therapy even in this subset of patients [20]. In the present study, we reported nearly 50% of pCR patients who did not receive adjuvant treatment, with a relative risk of death 3 times higher (95% CI: 1.302–6.809, *p* = 0.01) compared to patients who did.

However, the protective role of adjuvant therapy should be interpreted with caution. Indeed, considering the retrospective nature of the study, no established criteria exist based on which adjuvant therapy were proposed, despite the fact that a multidisciplinary tumor board discussed each single LA-NSCLC case before initiating the IT protocol, at restaging and after surgery, to evaluate the need of further therapies. It is logical to assume that a clinical selection may be the basis of such decision, which represents a remarkable bias. Likewise, it is similarly assumable that LA-NSCLC with a more advanced disease at presentation may be more frequently addressed to adjuvant therapy. Only focused randomized clinical trials may clarify the benefit of adjuvant therapy in this subset of patients, but hopefully, the advent of immunotherapy will completely revolutionize the overall treatment paradigm of resectable LA-NSCLC in a few years [25,26,27,28], as suggested by preliminary results [29,30].

Finally, we need to remark on the several limitations of the present study. Firstly, it is a retrospective study over a long time (more than 25 years) and a lot has evolved within the clinical path, such as diagnostic methods, surgical techniques, chemotherapeutic agents and radiotherapy techniques. Therefore, lots of concerns exist about whether the initial clinical stage is accurate or not. The staging might be under or overestimated, because not all patients underwent endobronchial ultrasound (EBUS), or mediastinoscopy or PET scan. Finally, while the overall mean follow-up duration was 56 months, a small percentage of cases had a short follow-up period (less than 12 months), which represents a further drawback. Overall, these biases might affect the results of this study, which is the main limitation of this study.

Therefore, in light of all limitations reported above, readers should interpret our results with caution, as multiple biases exist.

On the other hand, the present study has the merit to be focused on a specific population of NSCLC cases, collecting a large dataset, which is the main point of strength. We also identified prognostic factors in our cohort of patients, which could be taken into account when considering the opportunity to perform a mediastinal assessment in this subset of patients. However, only future prospective trials will give more appropriate information on this topic.

## 5. Conclusions

Despite the fact that multimodality treatments have been largely adopted in the last decades for LA-NSCLC, the survival benefit is almost uncertain, and the challenge remains to identify the best IT protocol able to obtain high response rates with acceptable associated toxicities. We observed that adjuvant therapy seems to be associated with better long-term survival, particularly in terms of distant disease control. Therefore, even in patients who achieved pCR after surgery following induction radio-chemotherapy, adjuvant therapy could be considered in the context of a clinical trial, especially in those patients with a higher relapse risk (i.e., baseline N2 multi-station involvement). However, large prospective controlled clinical trials should be completed before proposing adjuvant therapy as a daily clinical practice standard.

## Figures and Tables

**Figure 1 cancers-12-03572-f001:**
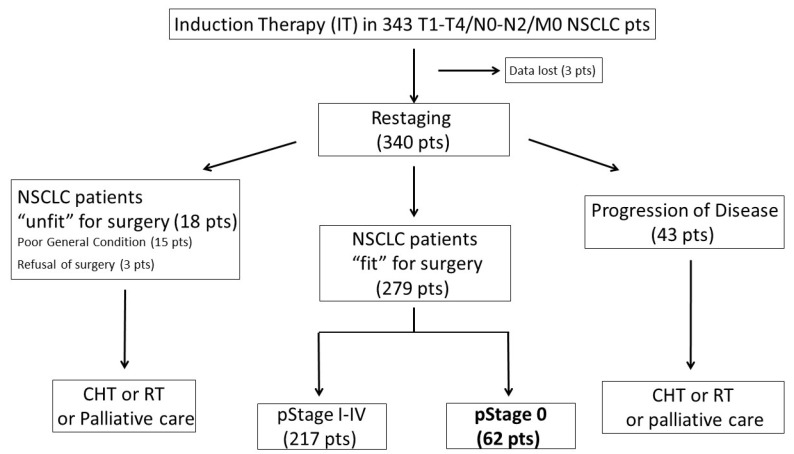
Consort-type diagram of the study group.

**Figure 2 cancers-12-03572-f002:**
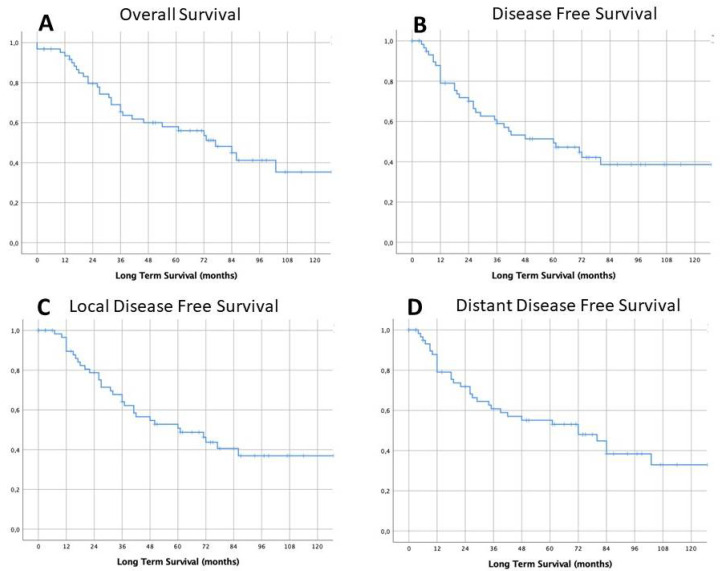
Survival long-term results: overall long-term survival (LTS) (**A**) and overall disease-free survival (DFS) (**B**), and local disease free survival (**C**) and distant disease-free survival (**D**).

**Figure 3 cancers-12-03572-f003:**
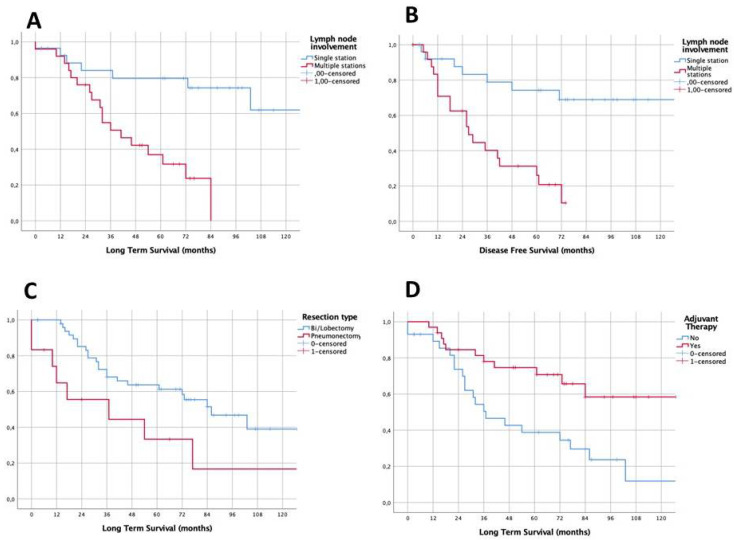
Survival functions according with nodal mediastinal involvement at diagnosis (**A**,**B**), surgical resection (**C**) and adjuvant treatment (**D**).

**Table 1 cancers-12-03572-t001:** Main sample clinical, surgical and pathological characteristics of the study group (*n* = 62).

Patients Characteristics	*n* (%)	Patients Characteristics	*n* (%)
**Sex (*n*(%))**		**Time between IT and surgery (average ± SD)**	54.91 ± 19.35
Female	5 (8.1%)		
Male	57 (91.9%)	**yTNM (*n* (%))**	
		0	9 (14.5%)
**Age (average ± SD)**	62.44 ± 8.76	IA	13 (20.9%)
		IB	0 (0%)
**cTNM (*n* (%))**		IIA	7 (11.3%)
IIB	3 (4.8%)	IIB	12 (19.4%)
IIIA	28 (45.2%)	IIIA	9 (14.5%)
IIIB	31 (50%)	IIIB	12 (19.4%)
**cT (*n* (%))**		**Kind of resection (*n* (%))**	
T1	5 (8.1%)	Lobectomy	46 (74.2%)
T2	17 (27.4%)	Pneumonectomy	12 (19.3%)
T3	25 (40.3%)	Bilobectomy	4 (6.5%)
T4	15 (24.2%)		
		**Histology (*n* (%))**	
cN (*n* (%))		Adenocarcinoma	30(48.4%)
N0	6 (9.7%)	Squamous cell carcinoma	32 (51.6%)
N1	3 (4.8%)		
N2	53 (85.5%)	**Removed nodes (average ± SD)**	9.07 ± 6.2
**Nodal involvement (*n*(%)) (*N* = 31)**		**Number of removed nodes (*n* (%))**	
Single station	31 (58.5%)	1–6	16 (25.8%)
Multiple station	25 (41.5%)	>6	46 (74.2%)
**NAD therapy (num of patients (%))**		**Morbidity (*n* (%))**	
**CHT**	11 (17.7%)		
**CHT/RT**	51 (82.3%)	No	45 (72.6%)
**NAD Protocols (num of patients (%))**			
**RT (50.4 Gy)**			
Standard: 180 cGy daily	16 (31.4%)		
Accelerated: 120 cGy twice daily	35 (68.6%)	Yes	17 (27.4%)
**CT**			
Carboplatin	7 (11.2%)	**Adjuvant therapy (*n*(%))**	
Cisplatin + Gemcitabine	46 (74.2%)	None	29 (46.8%)
5 FU−Cisplatin	9 (14.5%)	CT	22 (30.6%)
		RT	8 (9.7%)
**Toxicity (*n* (%))**		CT-RT	3 (12.9%)
No	44 (71%)		
Yes	18 (29%)		

**Table 2 cancers-12-03572-t002:** The pattern of failure after surgery following IT in patients with cPR.

Variations	Local Recurrence *n* (%)	Distant Recurrence *n* (%)
**First relapse site**		
Brain	-	14
Adrenal Gland	-	3
Liver	-	3
Controlateral Lung	-	2
Bone	-	1
Ipsilateral Lung	5	-
Mediastinal	2	-
Bronchial Stump	2	-
**Surgical Approach**		
Pneumonectomy (*n* = 12)	0 (0%)	6 (50%)
(Bi)Lobectomy (*n* = 50)	9 (18%)	17 (34%)
**Adjuvant Therapy**		
None (*n* = 29)	5 (17.2%)	15 (51.7%)
Chemotherapy alone (*n* = 22)	4 (18.2%)	6 (27.3%)
Radio-chemotherapy (*n* = 3)	0 (0%)	0 (0%)
Radiotherapy alone (*n* = 8)	0 (0%)	2 (25%)
**Total**	9 (14.5%)	23 (37.1%)

**Table 3 cancers-12-03572-t003:** Survival and DFS at 3 and 5 years from surgery; *p*-value of Log Rank Test for the comparison of survival curves.

Variation	Overall Survival			Disease-Free Survival		
	3 Years (Ns = 36)	5 Years (Ns = 28)		3 Years (Ndfs = 33)	5 Years (Ndfs = 25)	
	Survival%	Survival%	*p*-Value	Survival%	Survival%	*p*-Value
**Age**						
<60 years (*n* = 22)	52%	52%	0.280	43%	32%	0.07
60+ years (*n* = 40)	66%	60%		62%	54%	
**Sex**						
Females (*n* = 5)	57%	57%	0.697	60%	60%	0.754
Males (*n* = 57)	60%	56%		53%	44%	
**cStage**						
IIB (*n* = 3)	100%	100%	0.564	67%	67%	0.676
IIIA (*n* = 28)	58%	63%		55%	50%	
IIIB (*n* = 31)	58%	60%		51%	37%	
**cT**						
T1 (*n* = 5)	80%	80%	0.513	60%	36%	0.524
T2 (*n* = 17)	61%	61%		62%	62%	
T3 (*n* = 25)	62%	58%		48%	38%	
T4 (*n* = 15)	47%	36%		52%	40%	
**cN**						
N0 (*n* = 6)	83%	83%	0.038	67%	67%	0.026
N1 (*n* = 3)	0%	0%		0%	0%	
N2 (*n* = 53)	60%	55%		57%	46%	
**T-size**						
<50 mm (*n* = 27)	65%	60%	0.336	61%	48%	0.385
≥50 mm (*n* = 35)	56%	53%		47%	43%	
Nodal Involvement						
Single Station (*n* = 28)	80%	80%		79%	69%	
Multiple Stations (*n* = 25)	43%	32%	0.001	32%	19%]	0.001
**Num. of removed Lymph-Nodes (*n* (%))**						
1–6 (*n* = 16)	64%	59%		60%	41%	
>6 (*n* = 46)	63%	60%	0.404	54%	54%	0.555
**Resection Type**						
Pneumonectomy (*n* = 12)	44%	33%	0.024	29%	29%	0.220
(Bi)lobectomy (*n* = 50)	64%	61%		58%	48%	
**Histology**						
Adenocarcinoma (*n* = 30)	54%	44%	0.440	48%	38%	0.349
Squamous Cell Carcinoma (*n* = 32)	66%	66%		60%	52%	
**Adjuvant Therapy**						
No (*n* = 29)	43%	39%	0.005	34%	29%	0.013
Yes (*n* = 33)	75%	70%		69%	57%	
**Adjuvant Therapy Type**						
None (*n* = 27)	38%	34%	0.002	28%	23%	0.004
RT alone (*n* = 2) or CHT alone (*n* = 32)	77%	73%		71%	59%	
RT-CHT (*n* = 3)	67%	67%		67%	67%	
**Recurrence**						
No (*n* = 30)	93%	93%	0.000			
Yes (*n* = 32)	34%	28%				

Ns: number of survivors; Ndfs: number of disease-free survivors.

**Table 4 cancers-12-03572-t004:** Hazard ratios (HR [95% confidence interval (CI)]) for mortality and disease recurrence; *p*-value from Cox Regression.

Variables	Mortality		Disease Recurrence	
	HR (95% CI)	*p*-Value	HR (95% CI)	*p*-Value
**Type of resection**				
Pneumonectomy	2.613	0.036		
	(1.066; 6.407)			
(Bi)lobectomy	1			
**Adjuvant Therapy**				
No	2.98	0.01	2.607	0.041
	(1.302; 6.809)		(1.039; 6.542)	
Yes	1		1	
**N2-multistation disease**	1		1	
No				
Yes	4.37 (1.632; 11.695)	0.003	6.241 (2.305; 16.897)	<<0.001
**Age > 60 years** No			1	<<0.001
Yes			0.189	
			(0.074, 0.483)	

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
