# Peer review of "Prognostic Factors and Long-Term Survival in Locally Advanced NSCLC with Pathological Complete Response after Surgical Resection Following Neoadjuvant Therapy"

_cancers, 2020, doi:10.3390/cancers12123572_

Round 1
Reviewer 1 Report
I would like to thank authors for the effort made in anwering to all my doubts regarding this manuscript. I believe that the corrections introducted have improved the article quality.
However, it still remain some typiing mistakes that should be corrected.
Reviewer 2 Report
The reviewer's comments were well revised.
This manuscript is a resubmission of an earlier submission. The following is a list of the peer review reports and author responses from that submission.
Round 1
Reviewer 1 Report
Article: Prognostic factors and long-term survivals in locally advanced NSCLC with pathological complete response after surgical resection following neoadjuvant therapy.
Comment:
Authors reported a well conducted analysis on a relatively large cohort of locally advanced NSCLC patients who achieved a pathological complete response after induction therapy followed by surgery, focusing in the prognostic factors affecting long-term survivals. The manuscript is innovative and highlights some interesting aspects of this subset of patients which number is expected to rise with the introduction of immunotherapy. Statistical analysis is well conducted and appropriate to the aim of the study. However, there are some concerns that should be addressed.
Authors should describe their policy in avoiding any invasive mediastinal re-staging despite the high rate (45%) of clinical stage IIB, IIIA and IIB after IT. Regarding the abovementioned data, authors reported “Among pCR patients who underwent PET/CT scan (# 28 cases) a persistence of a significant uptake at the level of lung lesion 138 (# 9 cases) or mediastinal lymph nodes (#4 cases) was detected” but in the table 1 it seems that there are slightly more patients with persistent clinical LA-NSCLC.
The latest NCCN guidelines reported that the presence of N2 positive lymph nodes substantially increases the likelihood of positive N3 lymph nodes that could explain the high rate of recurrence. Authors should take into account the initial invasive mediastinal staging as prognostic factor.
One of the historical most controversial aspect is the use of Radiation therapy in the IT setting due to high risk of complication rate. Authors should explore this aspect, especially for patients who underwent pneumonectomy whose survival is almost 50% after 1 year (as can be seen in the K-M curve).
Authors should describe more in detail all the criteria for address patients to adjuvant therapy after surgery despite the ypT0N0.
In table 3 the sum of patients who underwent lymphadenectomy is 58 (1-6: 26, >6: 32) out of 62 patients underwent surgery, please correct.
In the text there are several typing errors that should be corrected.
Minor revisions:
Authors should revise the use of commas and points in the decimal number.
Line 21: IT should be defined the first time it’s used.
Line 62: Please correct the firm of the NCCN version in February 3,2020 or with the version. 2020
Line 91: 5-yrs survival should be corrected in 5-year survival.
Line 160: Windows Mac should be corrected.
Reviewer 2 Report
This is a retrospective analysis of patients who achieved pathological CR after induction chemoradiotherapy in locally advanced NSCLC. The author found that long-term results can be expected in patients with pCR especially in patients with single N2 station. This study is quite interesting but several issues should be concerned.
- This study was retrospective analyzed from 1992 to 2019, which is quite lon period time. During the 27 years, lots of treatment strategies have been evolved such as diagnositc method, surgical technique, chemotherapeutic agents and radiotherapy technique. Therefore, lots of concern exist whether the initial clinical stage is accurate or not. The staging might be under or overestimated, because not all patients underwent EBUS, or mediastinoscopy or PET scan. These might affect the results of this study, which is the main limitation of this study.
- The author analyzed only 62 patients who achieved pCR. It would be more valuable if the author compare the clinical results between 217 patients who did not achieve pCR and 62 pts. Also, patients with down stage (not pCR) have good prognosis, so these patients group should be analyzed and compared
- The author should describe in detail how the clinical stage was performed.
- The author should describe in detail whether the surgical resection technique was standardized among patients in terms of lymph node dissection number and method, and etc.
- The author should describe in detail whether the surveilance was unified for each patient in terms of follow-up period , diagnostic method , follow-up interval and etc.
- For surgical patients, at least 3 yr follow-up is required to analyzed DFS, so patients recruited from 2018 might be irrelevant to include in this analysis